# [RE] Background-Aware Pooling and Noise-Aware Loss for Weakly-Supervised Semantic Segmentation

## 1 Reproducibility Summary

*The following paper is a reproducibility report for **Background-Aware Pooling and Noise-Aware Loss for Weakly-Supervised Semantic Segmentation** [12] published in CVPR 2021 as part of the ML Reproducibility Challenge 2021. The repository is available at this link with the PyTorch Lightning [4] Extension available at this link.*

**Scope of Reproducibility**

The paper's central claim revolves around the newly introduced Background Aware Pooling method to generate high-quality pseudo labels using bounding boxes as supervision and Noise Aware Loss to train a segmentation network using those noisy labels. The authors assert that these two techniques combined set the new state-of-the-art for weakly supervised semantic segmentation on PASCAL VOC 2012 [3].

**Methodology**

We started with the publicly available code-base provided by the authors and reproduced the results associated with Stages 1 and 2 involving pseudo label generation. Further, we implemented NAL for Stage 3 training and used it to train a semantic segmentation network, reproducing its claims. We performed many refactoring and upgrades on the author's code to include various procedures mentioned in the paper.

**Results**

We reproduced and verified all the central claims made by the authors in the paper, confirming the intuition behind the novel methodologies introduced in the paper. Our results differ using the parameters given in the paper for the segmentation experiments but still support the claim of NAL being superior to its counterpart losses.

**What was easy**

The completed code for training the classification network and pseudo label generation using BAP was available in the authors' code-base, and the results associated with them were straightforward to reproduce.

**What was difficult**

Implementing some parts of Stage 1 and Stage 2 and the complete Stage 3 code, including NAL and further experimenting with them to resolve the minute issues, was the most challenging part of the reproduction. Even though authors gave detailed feedback, VOC-to-COCO conversion for unseen classes also posed many challenges.

**Communication with original authors**

Contact with authors was made via Email regarding specifications in methodologies involving pseudo label generation and VOC-to-COCO experiments. Apart from the code, comprehensive and helpful replies were given by them.

# 1   Introduction

Semantic segmentation, which is the pixel-wise classification of objects in images, finds crucial applications in areas such as autonomous driving, medical imaging, and augmented reality, to name a few. Training deep neural networks to perform this task accurately requires extensive and quality training data and annotating it, which is laborious and intensive. Weakly-supervised semantic segmentation (WSSS) techniques aim to ease the task of annotation by using image-level labels or object bounding boxes as a weak form of supervisory signal to generate possibly noisy "pseudo-ground-truth labels." While existing methods come at the expense of additional overheads, WSSS using background-aware pooling (BAP), introduces a technique to discriminate foreground and background regions within bounding boxes to generate quality pseudo labels at negligible overhead. On the other hand, Noise-Aware Loss (NAL) improves the performance of models by lessening the effect of incorrect pseudo labels during training.

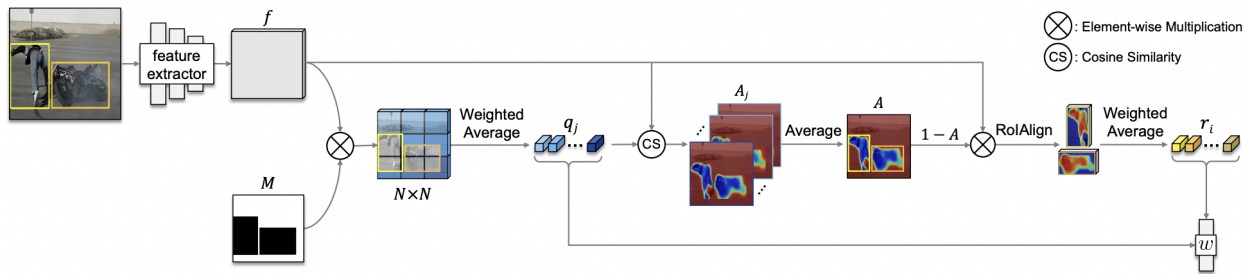

Figure 1: Image classification with Background Aware Pooling.

# 2   Scope of reproducibility

The paper introduces a new weakly supervised semantic segmentation technique using bounding box annotations to generate pseudo labels and train a segmentation network using those labels as supervisors.

Here are the major claims, summarized as follows:

1. High-quality pseudo segmentation labels are generated with the proposed Background Aware Pooling method using bounding box annotations in comparison to the conventional Global Average Pooling method [10, 18].

2. The novel Noise Aware Loss can use the unreliable regions present in the noisy pseudo labels.

3. Fully trained classification and Segmentation networks achieved the current state-of-the-art performance for weakly-supervised semantic segmentation on PASCAL VOC data-set using the above-presented methods.

# 3   Methodology

The main experiments of the paper are divided into three stages, as shown below:

1. Training a classifier network using Background-Aware Pooling (BAP) on the VOC dataset.

2. Generation and Evaluation of Pseudo labels generated on VOC for a model trained using BAP.

3. Training and evaluation of a model using Noise-Aware Loss (NAL) on the pseudo labels generated in Stage 2.

## 3.1   Method Descriptions

### 3.1.1   BAP in the training of Classification Network

The task of discriminating the foreground and background regions within a bounding box is approached as a retrieval task. Firstly, the feature map $f$ obtained from the model is divided into $N$ x $N$ regular grids denoted by $G(j)$. For each $G(j)$, features are aggregated as per Eq. (1) and are used as queries $q_j$ for the retrieval of background features

within each bounding box. For this purpose, a binary mask $M$ is defined, where for a position $p$ within a bounding box $M(\mathbf{p}) = 0$, and one otherwise.

$$q_j = \frac{\sum_{\mathbf{p} \in G(j)} M(\mathbf{p}) f(\mathbf{p})}{\sum_{\mathbf{p} \in G(j)} M(\mathbf{p})} \tag{1}$$

For a given grid cell $G(j)$, the term $A(j)$ is computed as shown by Eq. (2). Upon averaging overall $A_j(p)$, attention map, $A$ is obtained, corresponding to the likelihood that a given pixel belongs to the background. This is represented by Eq. (2), where $J$ denotes the total number of valid grid cells.

$$A(\mathbf{p}) = \frac{1}{J} \sum_j A_j(\mathbf{p}), \quad \text{where } A_j(\mathbf{p}) = \begin{cases} \text{ReLU} \left( \frac{f(\mathbf{p})}{\|f(\mathbf{p})\|} \cdot \frac{q_j}{\|q_j\|} \right) & , \mathbf{p} \in \mathcal{B} \\ 1 & , \mathbf{p} \notin \mathcal{B} \end{cases} \tag{2}$$

For a given bounding box $B_i$, foreground features $r_i$ are aggregated using the attention map $A(p)$ by means of a weighted average pooling, as per Eq (3). The authors refer to this process as Background-Aware Pooling (BAP). Finally, the $(L + 1)$ - way softmax classifier $w$ is applied to $r_i$ and $q_j$ corresponding to the foreground and background features, respectively, to train the model using standard cross-entropy loss.

$$r_i = \frac{\sum_{\mathbf{p} \in B_i} (1 - A(\mathbf{p})) f(\mathbf{p})}{\sum_{\mathbf{p} \in B_i} (1 - A(\mathbf{p}))} \tag{3}$$

### 3.1.2 Generation of Pseudo Labels

Two pseudo ground-truth labels namely $Y_{crf}$ and $Y_{ret}$ are generated from two complementary approaches. The first method involves using the background attention map and class activation maps (CAMs) [18] obtained from the classification network, and using them as the unary term for DenseCRF [8, 9, 15, 17]. The unary term for the background $u_0$ and unary term for object class $c$ denoted by $u_c$, is computed as shown in Eq. (4) and Eq. (5). The terms $u_0$ and $u_c$ for each class $c$ are then concatenated and provided as the unary term for DenseCRF to obtain $Y_{crf}$. Here $\mathcal{B}_c$ denotes the regions within bounding box(es) for class $c$ and $w_c$ is the classifier weight for object class $c$.

$$u_0(\mathbf{p}) = A(\mathbf{p}) \tag{4}$$

$$u_c(\mathbf{p}) = \begin{cases} \frac{\text{CAM}_c(\mathbf{p})}{\max_{\mathbf{p}}(\text{CAM}_c(\mathbf{p}))} & , \mathbf{p} \in \mathcal{B}_c \\ 0 & , \mathbf{p} \notin \mathcal{B}_c \end{cases}, \quad \text{where } \text{CAM}_c(\mathbf{p}) = \text{ReLU} \left( f(\mathbf{p}) \cdot w_c \right). \tag{5}$$

Generation of $Y_{ret}$, on the other hand, involves capturing the high-level features obtained from the classifier. Queries $q_c$ corresponding to prototypical features for each class $c$ is computed as per Eq. (7), where $\mathcal{Q}_c$ is the set of regions in $Y_{crf}$ labelled as class c. Following this, the correlation map $C_c$ for each class $c$ is shown below.

$$q_c = \frac{1}{|\mathcal{Q}_c|} \sum_{\mathbf{p} \in \mathcal{Q}_c} f(\mathbf{p}), \quad \text{and } C_c(\mathbf{p}) = \frac{f(\mathbf{p})}{\|f(\mathbf{p})\|} \cdot \frac{q_c}{\|q_c\|}. \tag{6}$$

However, the authors have applied the ReLU function over the mentioned cosine similarity in their official implementation. Finally, the argmax function is applied over the correlation map $C_c$ to obtain pseudo labels $Y_{ret}$.

**Pseudo-labels for Unseen Classes: "VOC-to-COCO"** The authors mention in the paper that their pseudo label generator is generic in that for classes unseen during training, $1 - u_0$ can be used as a class agnostic foreground attention map in place of the attention map obtained using the corresponding CAM.

We illustrate this by providing a generalized form of Eq (7) below.

$$u_c(\mathbf{p}) = \begin{cases} \frac{\text{CAM}_c(\mathbf{p})}{\max_{\mathbf{p}}(\text{CAM}_c(\mathbf{p}))} & , \mathbf{c} \in \mathcal{C} \text{ and } \mathbf{p} \in \mathcal{B}_c \\ 1 - u_0(p) & , \mathbf{c} \notin \mathcal{C} \text{ and } \mathbf{p} \in \mathcal{B}_c \\ 0 & , \mathbf{p} \notin \mathcal{B}_c \end{cases} \tag{7}$$

Where $\mathcal{C}$ represents the set of classes whose classifier weights are available with the generator, and $u_0$ corresponds to the background attention map attained in Eq. (5).

### 3.1.3 Noise-Aware Loss for Semantic Segmentation with Noisy Labels

The authors use Noise-Aware Loss to train DeepLab [2] models using $Y_{\text{crf}}$ and $Y_{\text{ret}}$ . Feature map $\phi$ is extracted from the backbone network and probability map $Y_{\text{pred}}$ is obtained by passing feature map $\phi$ through the forward classifier. Probability map $H$ is obtained by passing $Y_{\text{pred}}$ through Softmax classifier $W$. The authors denote the regions where both $Y_{\text{crf}}$ and $Y_{\text{ret}}$ give the same label as $\mathcal{S}$ and where both give different labels as $\sim \mathcal{S}$. For the confident regions $\mathcal{S}$, $ce$ loss is calculated using Eq. (8).

$$\mathcal{L}_{\text{ce}} = -\frac{1}{\sum_c |\mathcal{S}_c|} \sum_c \sum_{\mathbf{p} \in \mathcal{S}_c} \log H_c(\mathbf{p}), \tag{8}$$

Here $H_c$ is a probability for the class $c$ and $\mathcal{S}_c$ is the set of locations labeled as the class $c$ in $\mathcal{S}$. The unreliable regions $\sim \mathcal{S}$ cannot be ignored, and for determining the accuracy of the label prediction, $wce$ loss is proposed. For the loss computation, the authors build upon the assumption that the weights of the classifier network $W_c$ can be treated as a feature representing the corresponding class $c$. A correlation map $D_c$ is calculated per class using cosine similarity as a metric as described in Eq. (9).

$$D_c(\mathbf{p}) = 1 + \left( \frac{\phi(\mathbf{p})}{\|\phi(\mathbf{p})\|} \cdot \frac{W_c}{\|W_c\|} \right), \tag{9}$$

$$\sigma(\mathbf{p}) = \left( \frac{D_{c^*}(\mathbf{p})}{\max_c (D_c(\mathbf{p}))} \right)^{\gamma} \tag{10}$$

A confidence map is then calculated using Eq.(10). Here $c^*$ is obtained as $Y_{\text{crf}}$ labels corresponding to the respective class. $\gamma$ is a damping parameter that is always set greater than 1. The confidence map can predict the probability of each label being correct. Thus, $wce$ loss is calculated according to Eq.(11).

$$\mathcal{L}_{\text{wce}} = -\frac{1}{\sum_c \sum_{\mathbf{p} \in \sim \mathcal{S}_c} \sigma(\mathbf{p})} \sum_c \sum_{\mathbf{p} \in \sim \mathcal{S}_c} \sigma(\mathbf{p}) \log H_c(\mathbf{p}) \tag{11}$$

The final loss is calculated using Eq. (12), where $\lambda$ is a weighing parameter which balances $L_{\text{ce}}$ and $L_{\text{wce}}$ .

$$\mathcal{L} = \mathcal{L}_{\text{ce}} + \lambda \mathcal{L}_{\text{wce}} \tag{12}$$

## 3.2 Datasets

The primary dataset used in our experimentation is the PASCAL VOC 2012 containing 1464, 1449, and 1456 images in the train, val, and test split, respectively, of 21 object classes is used as the primary dataset to benchmark the proposed methods. An augmented dataset containing 10582 images was prepared using the technique described in [6] and used to train the classification and segmentation models. For a cross-dataset evaluation of the pseudo label generator, the train set of the MS COCO 2017 dataset [11] containing 117040 images (excluding grayscale images) of 81 classes are used.

## 3.3 Hyperparameters

Default hyper-parameters proposed in the paper were used for all the stages and are listed in Table 1. A hyper-parameter search was performed for the values of grid size, lambda, and damp parameters, results of which we report in Section 4.2.1 and Section 4.2.3 respectively.

| Stage 1 | | Stage 3 | |
|---|---|---|---|
| Hyper-parameter | Value | Hyper-parameter | Values (VGG [16] / ResNet [7]) |
| Grid Size | 4 | Dense CRF | (4, 121, 5, 3, 3) / (4, 67, 3, 3, 1) |
| ROI Size | (2, 2) | CS Classifier Temperature | 20 |
| Stage 2 | | Learning Rate | 1e-3 |
| Background Threshold | 0.99 | Gamma | 0.9 |
| Crop Size | (321, 321) | Step Size | 10 |
| DCRF | (4, 55, 3, 3, 3) | Lambda Weight | 0.1 |
| Grid Size | 1 | Damping Coefficient | 7 |

Table 1: Hyperparameters used all over the experiments.

## 3.4  Code details

The complete code containing the proposed NAL and all ablation studies both using PyTorch [13] and PyTorch Lightning along with WandB [1] integration is available at these links: (PyTorch, PyTorch Lightning). Links to all obtained pseudo labels and pre-trained models are also provided in README. Detailed discussion about the implementation is provided in the following sections.

### 3.4.1  Pseudo label generation from VOC to COCO

We perform a cross-dataset evaluation of pseudo generator on the MS COCO dataset for a model trained of PASCAL VOC. While the authors do not provide an implementation for the same, we implement the experiment from details provided in the paper and communication with the authors. We appropriately map the VOC classes to the corresponding classes in COCO using information available about both datasets to facilitate Eq (7). We follow standard protocols for evaluating the pseudo labels using the official COCO API.

### 3.4.2  Semantic segmentation with NAL

The original authors' code implementation contained Stage 1 and Stage 2, but the Stage 3 code was incomplete. We thus implemented the complete Stage 3 training from scratch, including the proposed NAL and the other loss functions discussed in section 4.2.2 based on the details from the paper. We train the model using cross-entropy loss and Noise Aware Loss and utilize the Polynomial LR Scheduler. Dense-CRF is also applied as post-processing as per the code provided in the authors' repository.

## 3.5  Computational requirements

The experiments have been performed on Google Colaboratory with NVIDIA Tesla K80 (NVIDIA-SMI 495.46, Driver Version: 418.67, CUDA Version: 11.2) and Kaggle cloud service platform with NVIDIA Tesla P100-PCIE-16GB (NVIDIA-SMI 495.46, Driver Version: 418.67, CUDA Version: 11.0). The time required for various experiments is mentioned in Table 2.

| Experiment performed | Backbone of the network | Time (in hours) |
|---|---|---|
| Stage-1 training | VGG-16 | 2.5 |
| Stage-2 pseudo label generation | VGG-16 | 0.5 |
| Stage-2 VOC to COCO conversion | VGG-16 | 54 |
| Stage-3 training with CRF or RET | VGG-16 | 7 |
| Stage-3 training with CRF or RET | ResNet-101 | 7.5 |
| Stage-3 training with NAL | VGG-16 | 10.5 |
| Stage-3 training with NAL | ResNet-101 | 12 |

Table 2: Approximate time required for each experiment conducted.

## 4  Results

We experimented and verified all the central claims made by the paper about BAP methodology and NAL on PASCAL VOC 2012 dataset. Following are the detailed description of the results obtained.

### 4.1  Results reproducing original paper

#### 4.1.1  Experiments with Background Aware Pooling

We successfully replicated the results reported in Table 3 from the original paper, and it supports claim 1 of BAP being a superior method to GAP presented in Section 2 .

| Method | Authors' Results | Our Results |
|---|---|---|
| GAP | 76.1 | 75.5 |
| BAP $Y_{crf}$ w/o $u_0$ | 77.8 | 77 |
| BAP $Y_{crf}$ | 79.2 | 78.8 |
| BAP $Y_{ret}$ | 69.9 | 69.9 |
| BAP $Y_{crf}$ & $Y_{ret}$ | 68.2 | 72.7 |

Table 3: Comparison of pseudo labels on the PASCAL VOC validation sets in terms of mIoU.

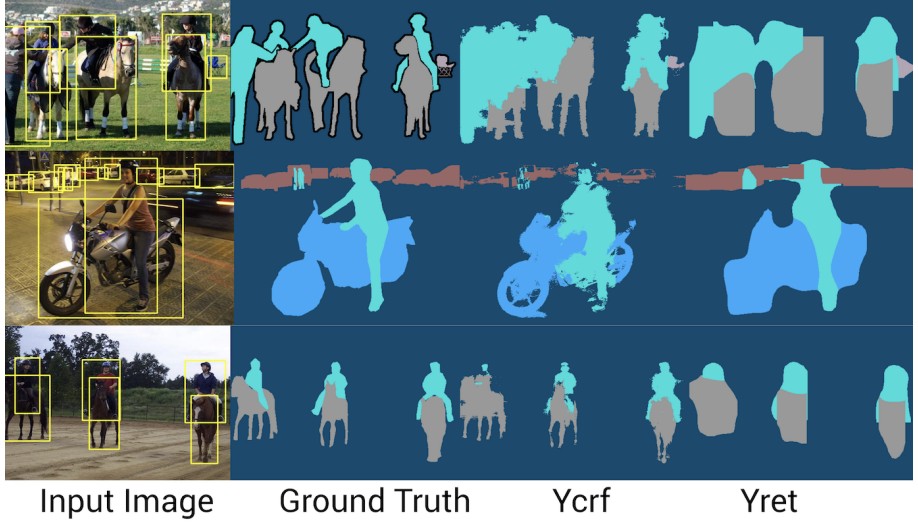

Input Image          Ground Truth          Ycrf          Yret

Figure 2: Visual examples of $Y_{crf}$, $Y_{ret}$ and the corresponding ground truth labels on PASCAL VOC validation set.

As discussed in section 3.1.2, we verified the authors' claims that the classifier model is generic and can be used for the detection of classes unseen during training. We trained the classifier model over the Pascal VOC dataset and generated pseudo labels over the MS-COCO dataset. We use the COCO-API evaluator of pycocotools to evaluate our results on the COCO benchmark. The comparison of our results with the authors' results is given in Table. 4.

| Method / Results | AP | $AP_{50}$ | $AP_{75}$ | $AP_S$ | $AP_M$ | $AP_L$ |
|---|---|---|---|---|---|---|
| BAP: $Y_{crf}$ $(Authors)$ | 11.7 | 28.7 | 8.0 | 3.0 | 15.0 | 27.1 |
| BAP: $Y_{crf}$ $(Ours)$ | 8.6 | 20.1 | 6.5 | 1.9 | 8.8 | 15.9 |
| BAP: $Y_{ret}$ $(Authors)$ | 9.0 | 30.1 | 2.8 | 4.4 | 10.2 | 16.2 |
| BAP: $Y_{ret}$ $(Ours)$ | 6.6 | 20.2 | 2.5 | 3.3 | 5.7 | 10.6 |

Table 4: Quantitative comparison of pseudo labels on the MS-COCO train set for model trained on Pascal VOC.

### 4.1.2 Experiments with Noise Aware Loss

Comparison between our and the authors' results regarding NAL is provided in Table 5, which shows that NAL outperforms the cross-entropy loss computed on $Y_{crf}$ and $Y_{ret}$, thus supporting the claim 2 presented in section 2.

| Method | DeepLab v1 | | DeepLab v2 | |
|---|---|---|---|---|
| | Author's Results | Our Results | Author's Results | Our Results |
| w / $Y_{crf}$ (*val*) | 67.8 | 64.7 | 74.0 | 67.0 |
| w / $Y_{ret}$ (*val*) | 66.1 | 62.8 | 72.4 | 70.2 |
| w / NAL (*val*) | 68.1 | 64.8 | 74.6 | 70.8 |
| w / NAL (*test*) | 69.4 | 65.6 | 76.1 | 71.7 |

Table 5: Comparison of mIoU scores using DeepLab-V1 and DeepLab-V2 on the PASCAL VOC 2012.

## 4.2 Results beyond original paper

### 4.2.1 Experiments with grid size

We performed a hyperparameter search for the grid size (N) and observed that lower values of N for generating pseudo labels provide the best results. In contrast, the opposite was true for training the classification network.

| Grid Size (N) | | For Generating | | |
|---|---|---|---|---|
| | | 1 | 2 | 3 |
| | 1 | 75.82 | 75.77 | 75.65 |
| | 2 | 76.11 | 76.10 | 75.15 |
| For Training | 3 | 75.87 | 75.78 | 75.81 |
| | 4 | **78.83** | 78.72 | 78.82 |
| | 5 | 74.16 | 74.07 | 74.02 |

Table 6: Comparison of our pseudo labels $Y_{crf}$ using different grid sizes on the PASCAL VOC val set.

### 4.2.2 Experiments with NAL and it's counterpart losses

Besides NAL, various other losses have been defined in the paper to deal with unreliable regions such as entropy regularisation and bootstrapping. The comparison between our results and the authors' results is given in Table 7, with both before and after applying Dense-CRF.

| Method | Authors' Results | Our Results |
|---|---|---|
| Baseline | 61.8 / 67.5 | 60.9 / 64.5 |
| w / Entropy Regularization [5] | 61.4 / 67.3 | 60.8 / 64.1 |
| w / Bootstrapping [14] | 61.9 / 67.6 | 60.9 / 64.6 |
| w / $\mathcal{L}_{wce}$ | 62.4 / 68.1 | 61.4 / 64.8 |

Table 7: Comparison of mIoU scores using different losses on the PASCAL VOC 2012 validation set.

### 4.2.3 Experiments with different values of lambda and damp parameters.

To justify the selection of the values of lambda and damp parameters, comparison studies were performed by choosing different values of lambda and damp parameters. We train the DeepLabV1 (LargeFOV) model for a range of lambda and damp parameters and report the results as a heat-map representation in Fig. 3.

| Damping Parameter | 0.05 | 0.5 | 0.7 |
|---|---|---|---|
| 70 | 0.6095 | 0.6074 | 0.6059 |
| 15 | 0.6109 | 0.6084 | 0.6045 |
| 3 | 0.6112 | 0.6073 | 0.6036 |

Lambda

Figure 3: mIoU scores obtained on the PASCAL VOC validation set.

## 5  Discussion

Through our experiments, we reproduce and verify the central claims of the original paper about the two newly introduced techniques - BAP and NAL. We additionally perform ablation studies on different model hyper-parameters and various losses to gain insights into the original author's choice of the same.

We obtained very similar results in the reproducibility of BAP. The above claim that BAP is a superior method to GAP is well verified by the increased results obtained using BAP compared to GAP on PASCAL VOC, as reported in Table 3. We further analyze that using $u_0$ (corresponding to background attention map) yields better results than using $u_b$ (corresponding to background class activation map) for generation of the pseudo labels, suggesting superior discrimination of background regions in this method.

In implementing the authors' cross-dataset evaluation results on the COCO dataset, we obtain considerably lower results despite following the protocols mentioned in the paper. However, our results support the claim that BAP serves as a promising technique in implementing a class-agnostic pseudo label generator.

We implemented NAL from scratch and performed all the weakly-supervised training experiments with the obtained pseudo labels $Y_{crf}$ and $Y_{ret}$. We report slightly lower results compared to authors, which we attribute to the minor implementational differences and a possible tuning of the parameters in DenseCRF. This can be shown by Table 7 in which all the results before DenseCRF match the author's results, but there are some differences after using DenseCRF. However, a relative gain in performance for both DeepLab v1 and v2 is clearly observed from Table 5 when unreliable regions are exploited with the help of NAL. Furthermore, our experiments using different losses for regions with different predicted labels in $Y_{crf}$ and $Y_{ret}$, as listed in Table 7, provide supporting evidence that NAL outperforms the contemporary losses and suggests it is a robust technique for weakly-supervised training when there are regions with less confidence.

For Stage 1 and Stage 2, we perform experiments with different choices of grid size in BAP, and for Stage 3, we analyze model performance for different values of damping parameter $\gamma$ and weighting parameter $\lambda$. From Table 6, we infer that the best result is obtained for grid size 4 for training and 1 for label generation, which is in coherence with the values used in the original paper. For Stage 3, Fig. 3 supports the authors' choice of values assigned to $\gamma$ and $\lambda$. Using a higher damping coefficient value ($\gamma$) makes the model biased towards most confident labels. On the other hand, using a higher value of $\lambda$ gives more weight to $wce$ loss, increasing the reliance on regions with low confidence. All the ablation experiments with the selected hyper-parameters yielded validation IoU lower than that obtained in Table 5.

In our qualitative analysis of the generated pseudo labels (refer Fig. (2)) $Y_{crf}$ and $Y_{ret}$ we infer that $Y_{crf}$ particularly performs well in capturing low level image features. In Fig. 2, it is seen to discriminate the background region between the wheel's spokes correctly. $Y_{ret}$, on the other hand, captures high-level features in the same image although mildly exaggerated. Thus, the two labels complement each other, and together is a good indication of unreliable regions identified and suppressed by NAL.

After porting the code base into PyTorch Lightning, we also concluded the implementations and experiments that ensured the correctness of various bits of training and evaluation process such as data loading, loss calculation, model weights optimization, and checkpoint re-loading for further reproducibility experiments in the future.

## 6  Conclusion

In this paper, we reproduce all the original results provided by the authors. Reproducing the first claim involving Background Aware Pooling, we were able to achieve similar results to the author. Hence, we support the claim that BAP is a superior method for WSSS than GAP. Cross dataset evaluation was performed on the COCO dataset. Our experiments verify the claim that the model works as a class agnostic pseudo label generator and achieves satisfactory results in performing VOC-to-COCO evaluation. For Stage 3, we implemented Noise Aware Loss from scratch and trained the DeepLab models for WSSS. Our results are slightly lower than the actual results. Nonetheless, our experiments still support the claim that NAL outperforms the contemporary losses and suggests it is a robust technique for weakly supervised learning. Our additional experiments also provide further insights on the performance of NAL for different values of hyperparameters. We thus believe it would be of interest to perform further experiments focused on modifying NAL, which might lead to better results.

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
