# OpenReview forum: "[RE] Background-Aware Pooling and Noise-Aware Loss for Weakly-Supervised Semantic Segmentation"
_ML_Reproducibility_Challenge/2021/Fall — RC2021_

### Official Review · Reviewer_RLe4 · 2022-03-17
**This RC paper is good but the original paper may not have enough impact**

**Rating:** 6
**Confidence:** 4

**Review:**


The paper shows that the authors have successfully implemented the paper "Background-Aware Pooling and Noise-Aware Loss for Weakly-Supervised Semantic Segmentation". The implementation illustrates that the BAP and NAL tricks for box-supervised semantic segmentation are effective.

My concern is the impact of "Background-Aware Pooling and Noise-Aware Loss for Weakly-Supervised Semantic Segmentation" is not significant enough. The original paper uses bounding box as supervision which is much stronger than image-level supervision which is widely used in other WSSS papers. Besides, the paper mainly reports the mIoU-based semantic results while it is more suitable to report the AP^{mask}-based instance segmentation results.

Nevertheless, this RC paper is good but the original paper may not have enough impact.

---

### Official Review · Reviewer_B1YL · 2022-03-27
**A study on Background-Aware Pooling and Noise-Aware Loss for Weakly-Supervised Semantic Segmentation**

**Rating:** 8
**Confidence:** 4

**Review:**

The reducibility summary is complete. There are some grammatical weaknesses, especially in comparison to the rest of the paper.

Instead of the scope of the reproducibility, the authors give a brief overview of the claims of the original paper. A more elaborate distinction between the original contributions and methods to investigate these claims would have been expected here.

The authors made use of the code the original authors provide with the paper. This code, however, is not complete and misses the third stage of the proposed method. The authors then proceeded to successfully implement the missing stage themselves. They also performed a hyperparameter search where they analyse the grid size, the lambda, and damp parameters. The written code is made available for further research.

Communication with the authors of the original paper was established via E-Mail and the authors state that helpful feedback was provided.

An ablation study was conducted to verify the original authors' design choices such as their newly presented Noise Aware Loss (NAL).

The reproducibility paper includes an elaborate discussion that summarises all conducted experiment and gives insights to the findings of the study. Despite having slightly worse results the authors still see the original claimed as verified and can show with their ablations that the design choices have a solid foundation.

In addition to the results that reproduce the original paper, the outcomes of several other experiments are shown and provide proof for the validity of the original paper.

Apart from a grammatically weaker beginning, the paper is well-written and shows the conduction of meaningful experiments.

---

### Meta-Review · Area_Chair_F6Sv · 2022-04-09

**Recommendation:** Accept
**Confidence:** 4

**Metareview:**

Reviewers praised that the authors reimplemented necessary parts of the original codebase, and conducted ablations, and the quality of the writing.

---

### Decision · Program_Chairs · 2022-04-09

**Decision:**

Accept

**Comment:**

Following the recommendation of reviewers and meta-reviewer, the paper is accepted for ML Reproducibility Challenge 2021, and will be published in the upcoming special edition of ReScience Journal.